# Amine Functionalization Leads to Enhanced Performance for Nickel- and Cobalt-Ferrite-Supported Palladium Catalysts in Nitrobenzene Hydrogenation

**DOI:** 10.3390/ijms241713347

**Published:** 2023-08-28

**Authors:** Viktória Hajdu, Ádám Prekob, Gábor Muránszky, Ferenc Kristály, Lajos Daróczi, Lajos Harasztosi, Zoltán Kaleta, Béla Viskolcz, Miklós Nagy, László Vanyorek

**Affiliations:** 1Institute of Chemistry, University of Miskolc, Miskolc-Egyetemváros, 3515 Miskolc, Hungary; viktoria.hajdu@uni-miskolc.hu (V.H.); adam.prekob@uni-miskolc.hu (Á.P.); gabor.muranszky@uni-miskolc.hu (G.M.); bela.viskolcz@uni-miskolc.hu (B.V.); 2Institute of Mineralogy and Geology, University of Miskolc, Miskolc-Egyetemváros, 3515 Miskolc, Hungary; askkf@uni-miskolc.hu; 3Department of Solid State Physics, University of Debrecen, 4010 Debrecen, Hungary; lajos.daroczi@science.unideb.hu (L.D.); lajos.harasztosi@science.unideb.hu (L.H.); 4Pro-Research Laboratory, Progressio Engineering Bureau Ltd., 8000 Szekesfehervar, Hungary; zoltan.kaleta@uni-miskolc.hu; 5Higher Education and Industrial Cooperation Centre, University of Miskolc, Miskolc-Egyetemváros, 3515 Miskolc, Hungary; 6Institute of Organic Chemistry, Semmelweis University, 1092 Budapest, Hungary

**Keywords:** hydrogenation, magnetic, stable catalyst, amine functionalized, aniline

## Abstract

Easy preparation, good yield and easy recovery are the key challenges in the development of industrial catalysts. To meet all these three criteria, we have prepared intelligent, magnetizable NiFe_2_O_4_- and CoFe_2_O_4_-supported palladium catalysts that can be easily and completely recovered from the reaction medium by magnetic separation. The fast and facile preparation was achieved by a solvothermal method followed by sonochemical-assisted decomposition of the palladium nanoparticles onto the surface of the magnetic nanoparticles. The metal–support interaction was enhanced by amine functionalization of the supports using monoethanolamine. The performance and stability of the non-functionalized and amine-functionalized NiFe_2_O_4_- and CoFe_2_O_4_-supported palladium catalysts were compared in the industrially important nitrobenzene hydrogenation reaction. All catalysts showed high catalytic activity during aniline synthesis; complete nitrobenzene conversion and high aniline yield (above 97 n/n%) and selectivity (above 98 n/n%) were achieved. However, during reuse tests, the activity of the non-functionalized catalysts decreased, as the palladium was leached from the surface of the support. On the other hand, in the case of their amine-functionalized counterparts, there was no decrease in activity, and a non-significant decrease in palladium content could be measured. Based on these results, it can be concluded that amine functionalization of transition metal ferrites may result in more effective catalysts due to the enhanced metal–carrier interaction between the support and the precious metal.

## 1. Introduction

Catalytic hydrogenation of nitro compounds is the main industrial method for the synthesis of suitable primary amines [1]. Catalytic hydrogenation of nitrobenzene is one of the most efficient ways to produce aniline, which is widely used in the production of agricultural chemicals, dyes, pharmaceuticals, biochemicals and polyurethanes [2,3,4]. A number of catalysts are used in the hydrogenation reaction, usually consisting of a catalytically active metal (e.g., Pd, Pt, Ni, Cu) and a support (e.g., carbon, zeolites, alumina, silica) [5,6,7,8,9].

Nanoparticles have emerged as effective alternative carrier materials due to their high specific surface area, excellent adsorption properties and good dispersibility in liquid media. However, the easy separation and recycling of nanoparticulate materials from the reaction medium remains a challenge [10]. In recent years, spinel ferrites (MFe_2_O_4_) have attracted the attention of many researchers due to their special properties, such as high electrical resistivity, high chemical/thermal stability, high permeability, ease of functionalization and magnetic strength [11,12,13]. Due to their properties, spinel ferrites play a major role in applications such as microwave and electronic devices [14], chemical sensors, data storage devices, magnetic resonance imaging (MRI) [15], drug delivery systems [16], and adsorption [17] and catalytic processes [18]. Unfortunately, spinel ferrite cannot form a stable dispersion system and the particles aggregate easily, but the functionalization of magnetic nanoparticles with different functional groups such as -COOH, -NH_2_ and -SH [19,20] may solve this problem. Functionalized magnetic nanoparticles are excellent alternatives to conventional materials as heterogeneous catalyst supports with high specific surface area [21,22]. Without surface modification, the hydrogenolysis products can induce deactivation of the supported precious metal catalysts owing to their detrimental poisoning effects. Consequently, a higher catalyst load is required to achieve the complete conversion of the reaction [23,24,25]. Ferrites have the additional advantage of being magnetically separable from the reaction medium, thus eliminating the need to centrifuge and filter the catalyst after the reaction is complete [26]. In industrial applications in particular, the catalyst’s durability and reusability are very important factors, along with its high catalytic activity.

Polshettiwar and co-workers [27] carried out a reuse test in the hydrogenation reaction of phenylacetylene using an amine-functionalized nanoferrite-supported Ni catalyst. The catalyst could be reused at least five times without any change in activity. The leaching of metals before and after the reaction was also investigated by inductively coupled plasma atomic emission spectroscopy (ICP-AES). The Ni concentration was 8.3% before and 8.2% after the reaction, which confirmed negligible leaching of Ni. Wang and co-workers [28] have investigated the reuse of thiol-modified magnetic nanoparticles (Fe_3_O_4_-SH-Pd) in the hydrogenation reaction of nitrobenzene. The results show that the catalyst can be used at least ten times without any change in activity. In addition, metal leaching was also studied. The Pd concentration was 5.3% before the reaction and 4.7% after the reaction, which represents negligible Pd leaching. Based on these results, it can be concluded that the functionalization of magnetic nanoparticles can provide a strong metal–carrier interaction, which can prevent the leaching of noble metals, thus increasing the stability of the catalyst.

In this work, we investigated the stability of non-functionalized and amine-functionalized NiFe_2_O_4_- and CoFe_2_O_4_-supported palladium catalysts in nitrobenzene hydrogenation reactions. The catalysts exhibit high activity and magnetic properties, which allow them to be easily separated from the reaction medium without loss.

## 2. Results

### 2.1. Characterization of the Amine-Functionalized and Non-Functionalized CoFe_2_O_4_ and NiFe_2_O_4_ Magnetic Catalyst Supports

The essence of the research is presented in Figure 1. Two different reaction routes, namely, solvothermal and sonochemical synthesis, were used to prepare amine- and non-functionalized cobalt and nickel ferrite nanoparticles. The nanoferrites serving as catalyst supports for magnetically separable intelligent hydrogenation catalysts were thoroughly investigated. We assumed that the presence of the NH_2_ groups on the catalyst surface would lead to enhanced stability and performance of the catalysts because of the stronger interaction between the Pd nanoparticles and the ferrite supports. Palladium deposition onto the ferrite nanoparticles was carried out by ultrahigh-energy ultrasound treatment in ethanolic media.

The specific surface area of the prepared ferrite samples was examined by CO_2_ adsorption–desorption measurements based on the Dubinin–Astakhov method. The highest surface area was measured in the case of the amine-functionalized cobalt ferrite (279.4 m^2^/g), much higher than that of its non-functionalized counterpart (18.2 m^2^/g). In the case of the amine-functionalized nickel ferrite, the results were 93.8 m^2^/g (amine-functionalized) versus 64.0 m^2^/g (non-functionalized).

The functional groups on the surface were identified by FTIR. The spectra of the amine-functionalized CoFe_2_O_4_ and NiFe_2_O_4_ samples revealed characteristic absorption bands of surface-bound amines and hydroxyl groups (Figure 2A). Between 1000 and 1100 cm^−1^, absorption of the νC–O and the νC–N vibrations (1060 cm^−1^, overlapping) are identified, which belong to the hydroxyl, carboxyl and amino groups, respectively. Another band, which is characteristic of the amino groups, was found between 1480 cm^−1^ and 1740 cm^−1^ which is a convolution of the νN–H and νC=C bands. The stretching vibration band of the N–H bonds is convoluted with the vibration band of the -OH groups in a broad peak in the 3000–3750 cm^−1^ region. The symmetric and asymmetric stretching vibrations of the aliphatic and aromatic C–H bonds resulted in absorption at 2877 cm^−1^ and 2935 cm^−1^, which can be explained by the adsorbed organic molecules (ethylene glycol, EG and monoethanolamine, MEA) on the surface [29,30]. Two characteristic peaks are shown at 421 cm^−1^ and 596 cm^−1^ wavenumbers, which were assigned as intrinsic stretching vibration modes of the metal-oxygen at the octahedral and tetrahedral sites [31].

In the spectra of the non-functionalized ferrite nanoparticles, fewer absorption bands are found than in the case of their amine-functionalized counterparts (Figure 2B). The band with low intensity at 1630 cm^−1^ may originate from the vibration of the adsorbed water molecules [32].

The morphology and particle size of the amine-functionalized and non-functionalized cobalt and nickel ferrite crystallites were investigated by HRTEM (Figure 3A,B and Appendix A). The HRTEM pictures of the NH_2_-functionalized ferrites clearly show the presence of globular aggregates of nanocrystals, which are composed of smaller, individual crystallites of the size 4–10 nm (Appendix A). The average size of the above-detailed individual ferrite crystallites was also calculated based on XRD measurements. The size of the individual crystallites, determined from the HRTEM pictures, are coherent with the XRD measurements. The average size of the crystallites, which build up the NiFe_2_O_4_-NH_2_ nanoparticles, is 6 ± 2 nm. In the case of CoFe_2_O_4_-NH_2_, the average crystallite size was 4 ± 2 nm.

In contrast, the average particle sizes of the CoFe_2_O_4_-NH_2_ and NiFe_2_O_4_-NH_2_ globular nanoaggregates based on HRTEM pictures are 50.9 ± 8.2 nm and 41.5 ± 12.9 nm, respectively (Table 1 and Appendix A). Significant differences in morphology and size distribution were observed in the case of the non-functionalized CoFe_2_O_4_ and NiFe_2_O_4_ samples (Figure 3D,E and Appendix A). The HRTEM images of the -NH_2_ free (non-functionalized) ferrites show individual, non-aggregated crystallites, which do not self-organize into spherical aggregates as observed in the case of the functionalized ferrites. However, although the average particle sizes of the CoFe_2_O_4_ and NiFe_2_O_4_ are similar (12.9 ± 4.8 and 11.2 ± 4.0 nm, respectively), they differ significantly from those of the amine-functionalized ferrites (Table 1). The “P90” and “P95” values stand for the mean diameter of 90% and 95% of the particles.

Selected area electron diffraction (SAED) measurements revealed that the nanoparticles on the HRTEM pictures were CoFe_2_O_4_ spinel (both AF and NF, Figure 3C,F) and NiFe_2_O_4_ spinel (Appendix A). The identification was carried out by correlating the measured d spacing with d-values in X-ray databases, based on the powder diffraction files of the CoFe_2_O_4_ (PDF 22-1086) and NiFe_2_O_4_ (PDF 10-0325) spinel structures.

The magnetization curve of the amine-functionalized CoFe_2_O_4_ samples was measured at 303 K using a vibrating-sample magnetometer (VSM). The magnetic saturation (Ms) reached 36 emu/g in the case of the CoFe_2_O_4_-NH_2_, as shown in Figure 4A. A similar Ms value (35 emu/g) was obtained in the case of the non-functionalized CoFe_2_O_4_ sample (Figure 4B). The magnetization curve of the CoFe_2_O_4_-NH_2_ shows a narrow hysteresis loop with low coercivity (Hc: 23.1 Oe) and low remanent magnetization (Mr: 1.3 emu/g), as can be seen in the inset of Figure 3A. The values of Hc and Mr were relatively small, which indicates the soft ferromagnetic nature of the synthesized particles at room temperature. Narrow hysteresis loops indicate that the sample can be easily demagnetized. But, in the case of the non-functionalized CoFe_2_O_4_, a much wider hysteresis loop is seen; the coercivity was 1364 Oe and the remanence (Mr) was 13.3 emu/g (Figure 3B), similar to the literature (Hc: 500–1500 Oe, Ms: 13–61 emu/g, Mr: 5–27 emu/g) [33]. In this sense, the non-functionalized CoFe_2_O_4_ sample contains semi-hard ferromagnetic particles.

In the case of the NiFe_2_O_4_-NH_2_ sample, the Ms was 39 emu/g and Mr was 0.7 emu/g, in addition to an 8.9 Oe coercivity, indicating soft ferromagnetic behavior (Figure 4C). The non-functionalized NiFe_2_O_4_ sample shows a slightly widened hysteresis loop, the Ms value was 26.7 emu/g, and the magnitudes of Mr (4.7 emu/g) and Hc (91.6 Oe) also indicated a soft ferromagnetic property (Figure 4D) [34,35,36,37].

### 2.2. Characterization of the Amine-Functionalized and Non-Functionalized Magnetic Pd/CoFe_2_O_4_ and Pd/NiFe_2_O_4_ Catalysts

For enhancing the catalytic activity of the magnetic nanoparticles, Pd was deposited on the surface using ultrasound. It is well accepted that amine groups bind strongly to platinum and palladium nanoparticles [38,39]. The decorated NPs were characterized by X-ray diffraction (XRD) measurements. In the case of the cobalt-containing spinel, reflexion peaks have been identified at 30.1° (200), 35.5° (211), 43.1° (220), 53.6° (312), 57.2° (303) and 62.7° (224) two theta degrees (PDF 22-1086), each of which corresponds to only one metal oxide phase, CoFe_2_O_4_, and thus, it is a pure cobalt ferrite sample (Figure 5A,B). This statement is valid for both samples produced by different synthesis methods. The (111) and (200) reflexions of the elemental palladium are located at 39.6° and 46.1° two theta degrees in the case of the amine-functionalized and the non-functionalized CoFe_2_O_4_ supported catalysts (PDF 46-1043).

On the diffractogram of the NiFe_2_O_4_-supported palladium catalysts, (111), (220), (311), (222), (400), (422), (511) and (440) reflexions were identified at 18.1°, 30.1°, 35.5°, 37.2°, 43.2°, 53.8°, 57.2° and 62.8° two theta degrees, which belong to the nickel ferrite spinel (PDF 10-0325). The pure spinel phase can only be detected in the case of the sample produced by coprecipitation (Figure 5C). In the case of the non-functionalized sample, other component was detected in addition to the spinel (Figure 5D). In the non-functionalized nickel ferrite sample, in addition to the spinel (66.3 wt%), nickel(II) oxide (NiO, bunsenite, 30.01 wt%) and FeNi_3_ (awaruite, 3.75 wt%) have also been identified. The presence of NiO has been confirmed by the peaks at 37.3° (111) and 43.2° (200) two theta degrees (PDF 47-1049). Furthermore, low intensity reflexions have been observed at 44.1° (111), 51.3° (200) and 62.9° (220) two theta degrees which correspond to FeNi_3_ (PDF 38-419).

The visual identification of palladium nanoparticles is difficult and uncertain on the ferrite nanoparticles due their small particle size. Therefore, high-angle annular dark-field (HAADF) pictures were taken and element mapping was carried out for the confirmation of their presence (Figure 6 and Appendix A). On the HAADF pictures, the Pd nanoparticles are shown as bright dots in contrast to the ferrite nanoparticles. Moreover, the element mapping confirmed the presence of the palladium particles in addition to the ferrites. Palladium nanoparticles (highlighted in yellow) are located on the surface of the spherical CoFe_2_O_4_ aggregates (Figure 6A,B) in the case of the amine-functionalized and the non-functionalized ferrites. Similar results were obtained in the case of the palladium-decorated nickel ferrite catalysts, too (Appendix A). The presence of cobalt and iron are also detectable and are also marked on the element maps.

### 2.3. Catalytic Performance of the Ferrite Supported Palladium Catalysts

First, the “bare” magnetic catalyst supports were tested in nitrobenzene hydrogenation. After four hours of hydrogenation, only 30.3 n/n%, 10.5 n/n% (CoFe_2_O_4_, CoFe_2_O_4_-NH_2_) and 35.7 n/n%, 14.5 n/n% (NiFe_2_O_4_, NiFe_2_O_4_-NH_2_) nitrobenzene conversions were achieved by using the cobalt and nickel ferrite supports, respectively. Interestingly, in the case of the non-functionalized ferrite supports, higher nitrobenzene conversions were obtained. Based on these results, we may conclude that the activity of the ferrite nanoparticles is not adequate for aniline synthesis. Next, the non-functionalized and amine-functionalized palladium-decorated ferrite catalysts were tested, and their activity was compared. Pd/CoFe_2_O_4_ resulted in slightly slower nitrobenzene conversion, since after 60 min 90.5 n/n% conversion was achieved, but in the case of its amine-functionalized counterpart 98.5 n/n% was measured at 323 K after one hour of hydrogenation (Figure 7A,C). After three hours of reaction time, an aniline yield above 98 n/n% was reached at 323 K in the case of the Pd/CoFe_2_O_4_ and Pd/CoFe_2_O_4_-NH_2_ (Figure 7B,D).

In the case of the Pd/NiFe_2_O_4_ and Pd/NiFeO_4_-NH_2_, the measured conversions were 97.7 n/n% and 65.8 n/n% after 60 min of hydrogenation at 323 K. It should be noted here that after 180 min the total amount of the nitrobenzene was hydrogenated for both catalysts (Appendix A). The maximum aniline yields were 97.2 n/n% (Pd/NiFe_2_O_4_) and 97.0 n/n% (Pd/NiFeO_4_-NH_2_) (Appendix A). In the case of the nickel-ferrite-supported catalysts, the non-functionalized sample showed higher catalytic activity with respect to conversion and yield.

Gas chromatography coupled with mass spectrometry (GC-MS) measurements revealed the presence of two intermediates, namely, azobenzene and azoxybenzene, which were not transformed to aniline at lower reaction temperatures (at 283 K and 293 K). One byproduct was formed near the end of the reaction, namely, N-methylaniline, because methanol (the reaction medium) methylates the amine functional group of aniline, which causes a small decrease in the aniline selectivity (Figure 8).

The non-functionalized and amine-functionalized catalysts were tested in four cycles at a temperature of 323 K and a 20 bar hydrogen pressure. As can be seen in Figure 9C,D and Appendix A, the results in the case of the amine-functionalized catalysts are almost identical for each reuse cycle (*n* = 4), i.e., there is no visible difference during the whole reaction time between the nitrobenzene conversions and aniline yields. The catalytic activity of the Pd/CoFe_2_O_4_-NH_2_ and Pd/NiFe_2_O_4_-NH_2_ remained constant and excellent even after repeated use. In contrast, it can be stated that the catalytic activity of the non-functionalized catalysts significantly decreases by the end of the fourth cycle (Figure 9A,B and Appendix A). The results suggest that amine-functionalized ferrite-supported palladium catalysts are more stable than their non-functionalized counterparts; no Pd loss occurred, possibly due to a strong interaction between the ferrite catalyst support and the palladium particles.

The results of ICP-OES measurements are in line with those of the reuse experiments, since the palladium content of the reused catalysts were very similar to that of the fresh (non-used) Pd/NiFe_2_O_4_-NH_2_ and Pd/CoFe_2_O_4_-NH_2_ samples (Table 2). This catalyst stability and the non-significant loss of the noble metal may be explained by the presence of the amine functional groups anchoring the Pd nanoparticles on the ferrite nanospheres. This theory is further supported by our previous results, where significant losses in catalytic activity and noble metals were observed in the case of non-functionalized ferrite supported catalysts.

## 3. Materials and Methods

### 3.1. Materials

The cobalt ferrite and nickel ferrite catalyst supports were synthetized from the following precursors: cobalt(II) nitrate hexahydrate, Co(NO_3_)_2_∙6H_2_O, MW: 291.03 g/mol (Sigma-Aldrich, Saint Louis, MO 63103, USA); nickel(II) nitrate hexahydrate, Ni(NO_3_)_2_∙6H_2_O, MW: 290.79 g/mol (Thermo Fisher GmbH, D-76870 Kandel, Germany); iron(III) nitrate nonahydrate, Fe(NO_3_)_3_∙9H_2_O (VWR International, Leuven, Belgium); ethylene glycol, HOCH_2_CH_2_OH, (VWR Int. Ltd., F-94126 Fontenay-sous-Bois, France); monoethanolamine, NH_2_CH_2_CH_2_OH (Merck KGaA, D-64271 Darmstadt, Germany); and sodium acetate, CH_3_COONa (ThermoFisher GmbH, D-76870 Kandel, Germany). Palladium(II) nitrate dihydrate, Pd(NO_3_)_2_·2H_2_O (Alfa Aesar Ltd., Ward Hill, MA 01835, USA) was used to deposit Pd onto the ferrite catalyst supports. Nitrogen (purity 4.0, Messer) and hydrogen (purity 4.0, Messer) were used during the experiments. Nitrobenzene (NB, Acros Organics, Morris Plains, NJ 07950, USA) was used as reactant during the catalytic hydrogenation tests. The analytical standards applied (azobenzene, nitrosobenzene, N-methylaniline) were purchased from Sigma-Aldrich Co. (St. Louis, MO 63118, USA).

### 3.2. Characterization Techniques

High-resolution transmission electron microscopy (Thermo Fisher Scientific Ltd., HRTEM, Talos F200X G2 electron microscope with field emission electron gun, X-FEG, accelerating voltage: 20–200 kV) was used for characterization of particle size and morphology of the ferrite and palladium nanoparticles. For the imaging and electron diffraction, a SmartCam digital search camera (Ceta 16 Mpixel, 4k × 4k CMOS camera) and high-angle annular dark-field (HAADF) detector were used. During sample preparation, the aqueous dispersion of ferrite was dropped on 300 mesh copper grids (Ted Pella Inc., 4595 Redding, CA 96003, USA). The phase analysis of the different oxide forms and their quantification was carried out with X-ray diffraction measurements using Rietveld analysis. A Bruker D8 diffractometer (Cu-Kα source) with parallel beam geometry (Göbel mirror) with a Vantec detector was applied. Average crystallite size of the domains was calculated by the mean column length calibration method using the full width at half maximum (FWHM) and the width of the Lorentzian component of the fitted profiles. Identification of the surface functional groups of the ferrite nanoparticles was carried out with Fourier transformed infrared spectroscopy (FTIR) by using the Bruker Vertex 70 equipment. During sample preparation, a 10 mg sample was pelletized with 250 mg spectroscopic-grade KBr; the measurements were performed in transmission mode. The specific surface area of the ferrite samples was examined by CO_2_ adsorption–desorption measurements at 273 K by using a Micromeritics ASAP 2020 sorptometer, based on the Dubinin–Astakhov method. The magnetic characterization of ferrite nanoparticles was carried out with a self-developed (by Harasztosi and Daróczi) vibrating-sample magnetometer (H&D 2000 VSM) system at room temperature. The magnetization (M) versus applied magnetic field (H) was performed over H up to 10,000 Oe.

### 3.3. Synthesis of the Amine-Functionalized Cobalt Ferrite and Nickel Ferrite Magnetic Catalyst Supports

Amine-functionalized CoFe_2_O_4_ and NiFe_2_O_4_ spinels were synthesized by using a modified coprecipitation method (Figure 1). In 150 mL (2.7 mol) ethylene glycol (EG), iron(III) nitrate nonahydrate and the chosen precursor (nickel(II) nitrate hexahydrate or cobalt(II) nitrate hexahydrate) were dissolved, as was sodium acetate (Table 3). The solution was heated to 100 °C in a round bottom flask under reflux with continuous stirring; after 30 min, 35 mL monoethanolamine (MEA, 0.58 mol) was added to this. After 12 h continuous agitation and reflux, the cooled solution was separated by centrifugation (4200 rpm, 10 min). The solid phase was washed with distilled water several times; the magnetic ferrite was easily separated by magnet from the aqueous phase. Finally, the ferrite samples were rinsed with anhydrous ethanol and were dried at 80 °C overnight. These ferrite-containing samples were used as magnetic catalyst supports for the preparation of palladium-decorated spinel catalysts.

### 3.4. Synthesis of the Non-Functionalized Cobalt Ferrite and Nickel Ferrite Magnetic Catalyst Supports

Cobalt- and nickel-containing ferrite nanoparticles were synthesized by using a two-step process which includes sonochemical treatment and combustion. In the first step, iron(III) nitrate nonahydrate and one of the precursors (Table 4) were dissolved in 20 g polyethylene glycol, and then the solutions were treated by using a Hielscher UIP1000 Hdt tip homogenizer for 3 min (130 W, 19 kHz).

### 3.5. Ultrasonication-Assisted Deposition of the Palladium Nanoparticles on Surface of the Cobalt and Nickel Ferrites

The non-functionalized and the amine-functionalized cobalt ferrites and nickel ferrites (2.00 g) were dispersed in 200 mL ethanol by ultrasonication with a Hielscher UIP1000 Hdt homogenizer. Into the ferrite dispersion was added 50 mL ethanolic solution of 0.25 g palladium(II) nitrate hydrate, and it was treated by ultrasonication for 5 min. The resulting Pd-decorated magnetizable ferrite catalysts were separated by magnets from the alcoholic phase, and they were washed with ethanol and dried at 378 K overnight.

### 3.6. Catalytic Tests of the Cobalt Ferrite and Nickel Ferrite Magnetic Catalysts

The hydrogenation of nitrobenzene in methanolic solution was carried out to study the catalytic activity of the magnetic-separable ferrite-supported palladium nanocomposites. The concentration of nitrobenzene was 0.25 mol·L^−1^, while 0.1 g catalyst was added to the system. The reaction took place in a Büchi Uster Picoclave reactor comprising a 200 mL stainless steel vessel equipped with a heating jacket. The pressure of hydrogen was kept at 20 bar, and the reactants were thermostated at 293 K, 303 K and 323 K. Sampling was carried out after 5, 10, 15, 20, 30, 40, 60, 80, 120, 180 and 240 min. After the hydrogenation studies, quantitative analysis of the samples was performed using an Agilent 7890A gas chromatograph coupled with Agilent 5975C mass-selective detector and RTX-624 column (60 m × 0.25 mm × 1.4 μm). The volume of the injected sample was 1 μL at a 200:1 split ratio, and the inlet temperature was set to 473 K. Helium was used as a carrier gas (2.28 mL/min) and the oven temperature was set at 323 K for 3 min, then heated to 523 K at a heating rate of 10 K/min and held for a further 3 min. Analytical standards for the main product, by-products and intermediates were obtained from Sigma Aldrich (Burlington, MA, USA) and Dr. Ehrenstorfer Ltd. (Wesel, Nordrhein-Westfalen, Germany). The efficiency of the catalyst was characterized by calculating the conversion (*X*%) of nitrobenzene based on the following equation (Equation (1)):(1)$$X\;\left(\%\right)=\frac{n_{\left(NB,\;consumed\right)}}{n_{(NB,\;initial})}\cdot 100$$

Aniline (*AN*) yield (*Y*%) was also calculated as follows (Equation (2)):(2)$$Y\;\left(\%\right)=\frac{n_{\left(AN,\;formed\right)}}{n_{\left(AN,\;theoretical\right)}}\cdot 100$$

Furthermore, *AN* selectivity (*S*%) was calculated according to the following equation (Equation (3)):(3)$$S\;\left(\%\right)=\frac{Y}{X}\cdot 100$$

## 4. Conclusions

Non-functionalized (NF) and amine-functionalized (AF) cobalt and nickel ferrite nanoparticles were synthetized, and their performance was compared in nitrobenzene hydrogenation reaction as bare catalyst supports and in palladium-decorated form. NF magnetic nanoparticles were of small particle size, 12.9 ± 4.8 (CoFe_2_O_4_) and 11.2 ± 4.0 nm (NiFe_2_O_4_), and cobalt ferrite only contained a spinel phase, whereas the nickel ferrite sample also contained bunsenite and awaruite in addition to nickel spinel. Contrary to their NF counterparts, the amine-functionalized cobalt and nickel ferrite magnetic catalyst supports turned out to be spherical aggregates (50.9 ± 8.2 nm for CoFe_2_O_4_-NH_2_ and 41.5 ± 12.9 nm for NiFe_2_O_4_-NH_2_) composed of smaller individual nanoparticles of the size 4–10 nm. Based on XRD measurements, the exclusive presence of spinel phases could be detected without any non-magnetic oxides. The advantage of the surface NH_2_-groups is that they promote the dispersibility of the particles in an aqueous medium and at the same time enhance the anchoring of precious metals to the surface of the nanoparticles.

To test this assumption, palladium nanoparticles were deposited onto the magnetic nanoparticles applying a fast and facile sonochemical method, yielding an immediately usable catalytically active form. Although total nitrobenzene conversion was achieved both in the case of the non-functionalized (Pd/NiFe_2_O_4_, Pd/CoFe_2_O_4_) and the amine-functionalized (Pd/NiFe_2_O_4_-NH_2_, Pd/CoFe_2_O_4_-NH_2_) catalysts at a 323 K hydrogenation temperature, significant differences in catalyst stability were observed. During the reuse tests of the amine-functionalized catalysts, the conversion and yield values remained virtually unchanged and excellent after four catalytic cycles. In contrast, the non-functionalized catalysts exhibited a significant decrease in their catalytic activity. ICP-OES measurements revealed that the loss of palladium from the surface of the catalyst supports is responsible for the loss in catalytic activity. In the case of AF nanoparticles, the catalyst stability and the non-significant loss of the noble metal may be explained by the amino functional groups anchoring the Pd nanoparticles on the ferrite nanospheres. In summary, amine functionalization of magnetic ferrite supports can significantly enhance the stability and therefore the reusability of the catalysts.

## Figures and Tables

**Figure 1 ijms-24-13347-f001:**
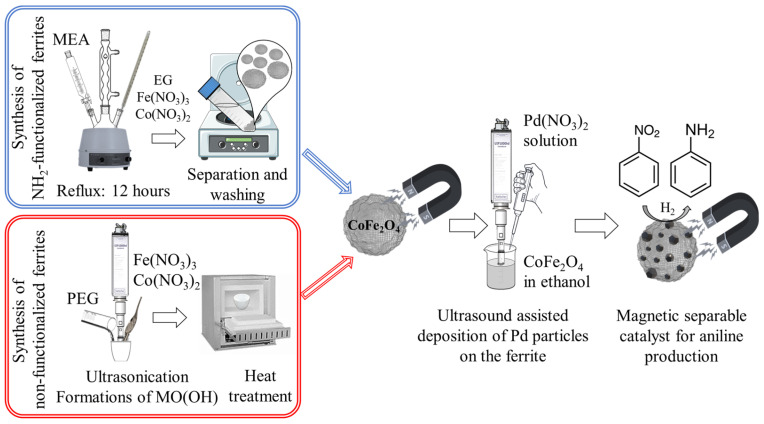
Synthesis of the non-functionalized and amine-functionalized magnetically separable ferrite-supported catalysts. The scheme is valid for nickel ferrites, too. MEA, EG and PEG stand for monoethanolamine, ethylene-glycol and poly(ethylene-glycol), respectively.

**Figure 2 ijms-24-13347-f002:**
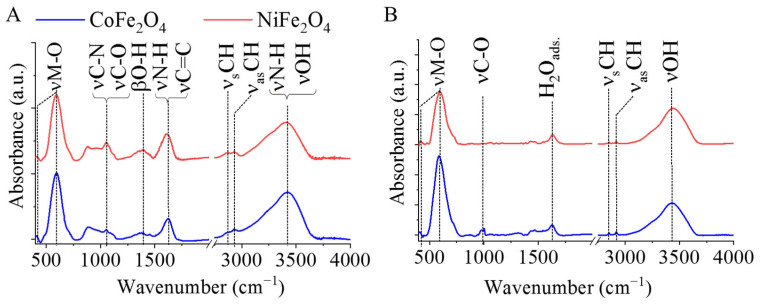
FTIR spectra of the amine-functionalized (**A**) and non-functionalized (**B**) CoFe_2_O_4_ and NiFe_2_O_4_ catalyst supports.

**Figure 3 ijms-24-13347-f003:**
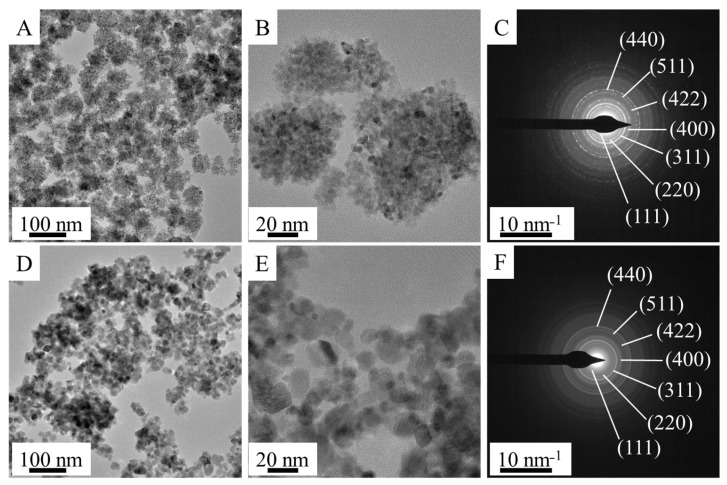
(**A**,**B**,**D**,**E**) HRTEM and (**C**,**F**) electron diffraction (selected area electron diffraction, SAED) pictures of the amine-functionalized (**A**–**C**) and non-functionalized (**D**–**F**) CoFe_2_O_4_ catalyst supports.

**Figure 4 ijms-24-13347-f004:**
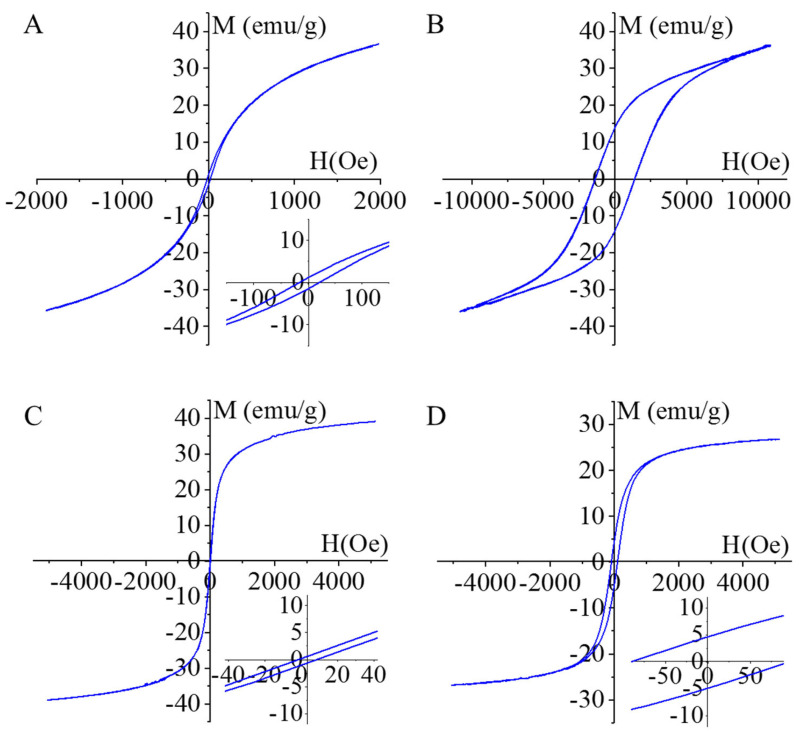
Magnetization curves of CoFe_2_O_4_-NH_2_ (**A**), CoFe_2_O_4_ (**B**), NiFe_2_O_4_-NH_2_ (**C**) and NiFe_2_O_4_ (**D**) samples.

**Figure 5 ijms-24-13347-f005:**
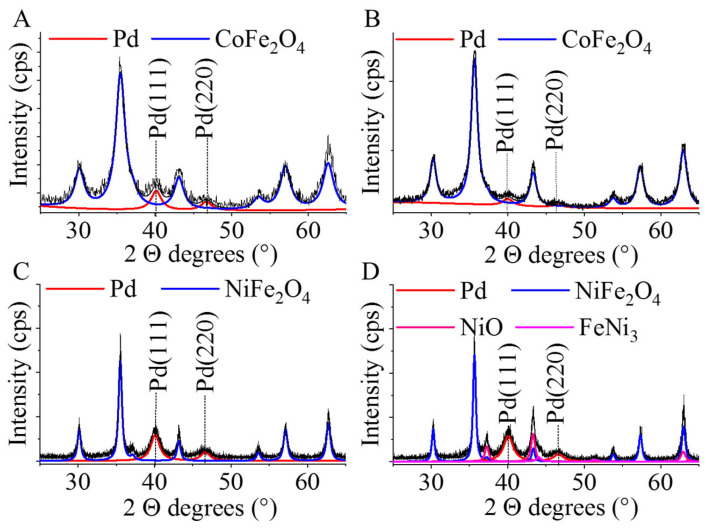
X-ray diffraction (XRD) patterns of the non-functionalized (**B**,**D**) and the amine-functionalized (**A**,**C**) cobalt-ferrite- and nickel-ferrite-supported palladium catalysts.

**Figure 6 ijms-24-13347-f006:**
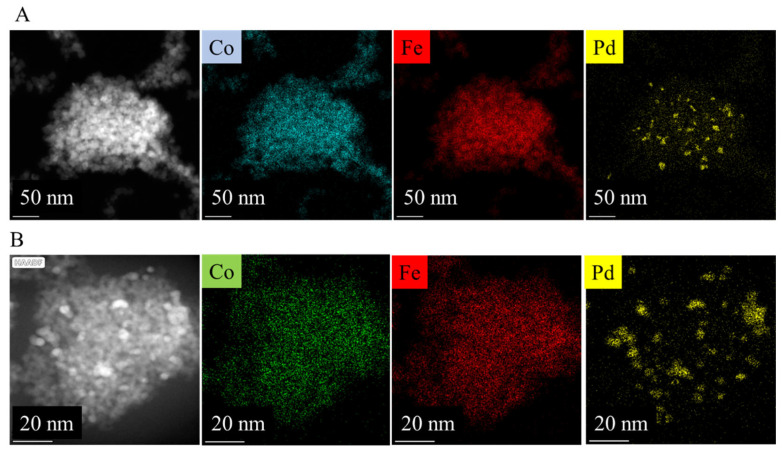
Element mapping of the amine-functionalized (**A**) and the non-functionalized (**B**) Pd/CoFe_2_O_4_ catalysts.

**Figure 7 ijms-24-13347-f007:**
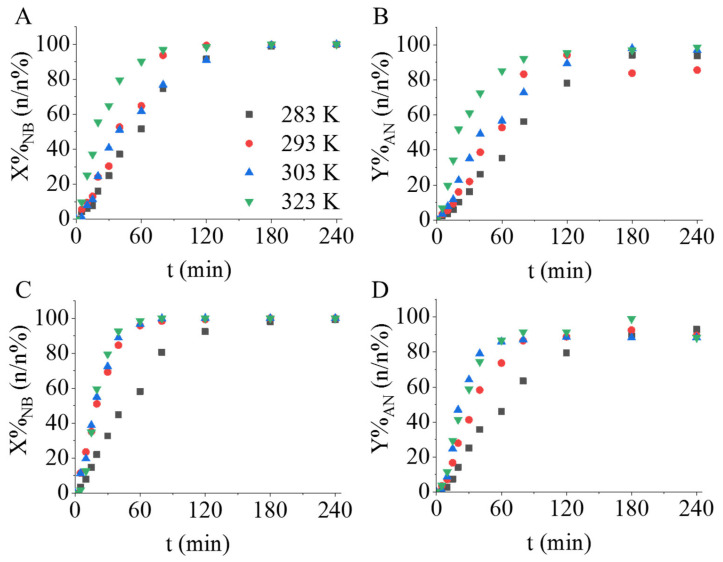
Nitrobenzene conversions and aniline yields versus time of nitrobenzene hydrogenation at 283 K, 293 K, 303 K and 323 K by using of Pd/CoFe_2_O_4_ (**A**,**B**) and Pd/CoFe_2_O_4_-NH_2_ (**C**,**D**) catalysts.

**Figure 8 ijms-24-13347-f008:**
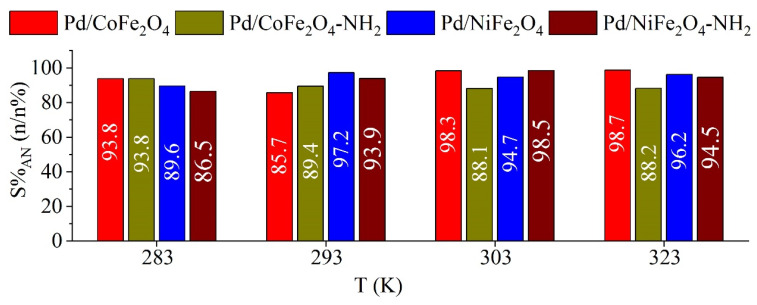
Selectivity of aniline at four reaction temperatures for the magnetic catalysts after 240 min of hydrogenation.

**Figure 9 ijms-24-13347-f009:**
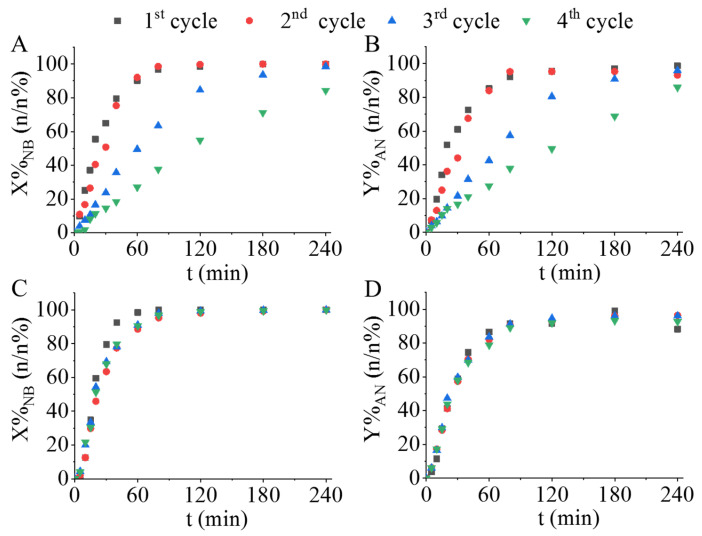
Reuse tests of the catalysts. Nitrobenzene conversion and aniline yield versus time of hydrogenation by using of Pd/CoFe_2_O_4_ (**A**,**B**) and Pd/CoFe_2_O_4_-NH_2_ (**C**,**D**) catalysts.

**Table 1 ijms-24-13347-t001:** Results of the size analysis (in nm) of the ferrite nanospheres of the amine-functionalized spinel samples and particles of the non-functionalized ferrites (based on HRTEM pictures).

	Mean	SD	Min.	Max.	P90	P95
Sample	Size (nm)					
CoFe_2_O_4_-NH_2_	50.9	8.2	28.8	66.0	61.2	62.6
CoFe_2_O_4_	12.9	4.8	3.6	26.9	18.5	22.4
NiFe_2_O_4_-NH_2_	41.5	12.9	20.5	71.2	61.6	66.4
NiFe_2_O_4_	11.2	4.0	4.4	23.2	17.2	18.4

**Table 2 ijms-24-13347-t002:** Results of the ICP analysis of the palladium content of the non-functionalized and amine-functionalized ferrite-supported catalysts before and after the reuse tests.

(wt%)	Before Use	After Four Uses
Pd/CoFe_2_O_4_-NH_2_	4.85	4.47
Pd/CoFe_2_O_4_	3.47	2.30
Pd/NiFe_2_O_4_-NH_2_	3.98	3.79
Pd/NiFe_2_O_4_	4.15	2.69

**Table 3 ijms-24-13347-t003:** Quantities (mol) of the reactants used during the preparation of the magnetic ferrite catalyst supports.

	Fe(NO_3_)_3_·9H_2_O	Ni(NO_3_)_2_·6H_2_O	Co(NO_3_)_2_·6H_2_O	CH_3_COONa
NiFe_2_O_4_-NH_2_	20 mmol	10 mmol	-	150 mmol
CoFe_2_O_4_-NH_2_		-	10 mmol	

**Table 4 ijms-24-13347-t004:** Quantities (mol) of the reactants used during the preparation of the magnetic ferrite catalyst supports.

	Fe(NO_3_)_3_·9 H_2_O	Ni(NO_3_)_2_·6 H_2_O	Co(NO_3_)_2_·6 H_2_O
NiFe_2_O_4_	14 mmol	7 mmol	-
CoFe_2_O_4_		-	7 mmol

## Data Availability

Data is available upon request from the corresponding authors.

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
