# Peer review of "Amine Functionalization Leads to Enhanced Performance for Nickel- and Cobalt-Ferrite-Supported Palladium Catalysts in Nitrobenzene Hydrogenation"

_ijms, 2023, doi:10.3390/ijms241713347_

Round 1

Reviewer 1 Report

Comments and suggestions for authors.

Author Response

Response to Reviewer 1 Comments

Thank You for Your positive comments! We appreciate it!

Comments and suggestions for authors.

An interesting result was obtained in the article - an increase in the stability of catalysts decorated with palladium nanoparticles after amination.

Point 1: How amination affects the electronic state of palladium? How is the interaction with amino groups carried out? The IR spectra of the samples do not give a clear view. It is necessary to know at least data on the amount of nitrogen on the surfaces.

Response 1: Thank You for this insightful comment!

Other groups carried out extensive investigations considering the interaction of NH2 groups and platinum and palladium.

The following text was inserted into line 187:

“It is well-accepted that amine groups bind strongly to platinum and palladium nanoparticles. [38-39]”

  1. Mandal, S.; Selvakannan, P. R.; Roy, D.; Chaudhari, R. V.; Sastry, M., A new method for the synthesis of hydrophobized, catalytically active Pt nanoparticles. Chemical Communications 2002, (24), 3002-3003. https://doi.org/10.1021/cm0352504
  2. Crooks, R. M.; Zhao, M.; Sun, L.; Chechik, V.; Yeung, L. K., Dendrimer-Encapsulated Metal Nanoparticles:  Synthesis, Characterization, and Applications to Catalysis. Accounts of Chemical Research 2000, 34, (3), 181-190. https://doi.org/10.1021/ar000110a

Point 2:  The presented data on catalytic transformations are not very clear for understanding: - Figure 7 shows a comparison of all catalysts in terms of selectivity; however, data on aniline yields are given only in fig. 8, from which it is difficult to compare the catalytic activity of different systems.

Response 2a:

  • Selectivity is the most clear representation of the catalyst efficacy, therefore it is presented individually. Since the value of yield can be derived from the NB conversion and AN selectivity values, presenting either the AN yield or selectivity is enough, but in some cases presenting both can be easier to understand.

- It follows from Fig. 9 that the yield of aniline on the Pd/CoFe2O4-NH2 sample in the 4th cycle is higher than in the first one, in contrast to the Pd/CoFe2O4 sample. What is the reason for this effect? On fig. 9 should also indicate the test temperature

Response 2b:

  • The inherent error of the GC measurements is about ±5%. Since the yield values of the different cycles are very similar it later cycles may appear to have higher yileds than previous ones. However, the differences are not significant and are withing the margin of error.

Point 3: In Fig.5 - the samples are incorrectly marked.

Response 3: Thank you the remark. We corrected the description of the figure.

Point 4: Table. 1 shows the values of P90 and P95. However, nowhere is it explained what it is.

Response 4: Thank you for your comment. The followint text was added into line 154: “The “P90” and “P95” values stand for the mean diameter of 90% and 95% of the particles.”

Point 5: Apparently, a typo was made in the conclusion - (line 373) - dinitrobenzene.

Response 5: Thank you the remark, we corrected the name in the text.

Point 6: The authors state in conclusion that “amine functionalization of magnetic ferrite supports can significantly enhance catalytic performance and reusability”

However, catalytic performance was practically unchanged or even decreased by amination, in contrast to reusability

Response 6: The line in question was rephrased as follows:

„In summary, amine functionalization of magnetic ferrite supports can significantly enhance the stability and therefore the reusability of the catalysts.”

Point 7: The authors refer to Supplementary materials, but they are not given.

Response 7: During manuscript submission there was no separate link for Supplementary  Material upload, so we followed the instructions and uploaded the manuscript and SI material in one compressed file. Since this issue happened to us before we double checked that the SI was uploaded.

We apologize for any inconvenience this issue may have caused. Next time we will consult with the editor to make sure the SI will be received by the referees.  

Reviewer 2 Report

The work submitted by Hajdu et al. is a really interesting and valuable study of enhanced performance of amine functionalization of nickel- and cobalt-ferrite supported palladium catalysts in nitrobenzene hydrogenation. The authors suggest really interesting and effective approach for the recovery of the catalysts by the magnetic separation from the reaction media. It was shown that amine functionalized catalysts are more stable against the Pd leaching comparing to the non-functionalized ones. It should be mentioned that the manuscript is well-written, the introduction clearly lays out reasons for studying of such system. The work has been carefully done and the results sound. But I would recommend the presented manuscript for publication in the IJMS only after the following issues are taken into account in a revised version.

Comments to authors:

- I could not find the Supplementary Information. Therefore it is really hard to evaluate the results which were obtained for NiFe2O4 based catalyst.

- Figure 5. It looks like something wrong with the Figure capture: A and B – non-functionalized, C and D – functionalized. But according to the legend A and B – CoFe2O4 based catalysts, C and D – NiFe2O4 based. It should be corrected.

- Some of the links do not meet the requirements of the journal. For example [16] - Srinivasan, S.Y.; Paknikar, K.M.; Bodas, D.; Gajbhiye, V. Applications of Cobalt Ferrite Nanoparticles in Biomedical Nanotechnology. https://doi.org/10.2217/nnm-2017-0379 2018, 13, 1221–1238, doi:10.2217/NNM-2017-0379. – journal name is missing.

Author Response

Response to Reviewer 2 Comments

Comments and Suggestions for Authors

The work submitted by Hajdu et al. is a really interesting and valuable study of enhanced performance of amine functionalization of nickel- and cobalt-ferrite supported palladium catalysts in nitrobenzene hydrogenation. The authors suggest really interesting and effective approach for the recovery of the catalysts by the magnetic separation from the reaction media. It was shown that amine functionalized catalysts are more stable against the Pd leaching comparing to the non-functionalized ones. It should be mentioned that the manuscript is well-written, the introduction clearly lays out reasons for studying of such system. The work has been carefully done and the results sound. But I would recommend the presented manuscript for publication in the IJMS only after the following issues are taken into account in a revised version.

Thank You for Your positive comments! We appreciate it!

Point 1: I could not find the Supplementary Information. Therefore it is really hard to evaluate the results which were obtained for NiFe2O4 based catalyst.

Response 1: During manuscript submission there was no separate link for Supplementary  Material upload, so we followed the instructions and uploaded the manuscript and SI material in one compressed file. Since this issue happened to us before we double checked that the SI was uploaded.

We apologize for any inconvenience this issue may have caused!

Next time we will consult with the editor to make sure the SI will be received by the referees.  

Point 2:  Figure 5. It looks like something wrong with the Figure capture: A and B – non-functionalized, C and D – functionalized. But according to the legend A and B – CoFe2O4 based catalysts, C and D – NiFe2O4 based. It should be corrected.

Response 2: Thank you the remark! Figure 5 label was modified as follows:

Figure 5. X-ray diffraction (XRD) patterns of the non-functionalized (B and D) and the amine-functionalized (A and C) cobalt ferrite and nickel ferrite supported palladium catalysts.

Point 3: Some of the links do not meet the requirements of the journal. For example [16] - Srinivasan, S.Y.; Paknikar, K.M.; Bodas, D.; Gajbhiye, V. Applications of Cobalt Ferrite Nanoparticles in Biomedical Nanotechnology. https://doi.org/10.2217/nnm-2017-0379 2018, 13, 1221–1238, doi:10.2217/NNM-2017-0379. – journal name is missing.

Response 3: Thank you the remark! The references have been checked and corrected accordingly.

Author Response

Response to Reviewer 3 Comments

Comments and Suggestions for Authors

The article „Amine functionalization leads to enhanced performance for 2 nickel- and cobalt-ferrite supported palladium catalysts in nitrobenzene hydrogenation” of authors Viktória Hajdu, Ádám Prekob, Gábor Muránszky, Ferenc Kristály, Lajos Daróczi, Lajos Harasztosi, Zoltán Kaleta, Béla Viskolcz, Miklós Nagy and László Vanyorek describes the effect of amine-based Pd anchoring in case of monoethanolamine impregnated ferrite on hydrogenation of nitrobenzene.

The authors compare structures of prepared mixed oxides metal-FeOx nanoparticles (some of them modified by MEA) and the same materials after Pd-anchoring using XRD, HRTEM, FTIR and detected formation of spinel-like structures. The authors effectively applied ICP-AES technique for determination of Pd lost from above-mentioned materials used in hydrogenation procedure.

The obtained results documented both interesting catalytic activity and long-term stability of newly prepared NH2-modified materials.

I mean that this article should be attractive for „International Journal of Molecular Sciences“ readers interested in new trends in heterogeneous catalysts chemistry.

The major drawback of this manuscript deals with missing data.

Thank You for Your positive comments! We appreciate it!

Authors refer to supporting information (S1-S6) in text of their manuscript. Unfortunately, no “Supplementary Materials” section appears in submission system.

During manuscript submission there was no separate link for Supplementary  Material upload, so we followed the instructions and uploaded the manuscript and SI material in one compressed file. Since this issue happened to us before we double checked that the SI was uploaded.

We apologize for any inconvenience this issue may have caused. Next time we will consult with the editor to make sure the SI will be received by the referees.  

Point 1: Please, unify the nomenclature for monoethanolamine, see for example differences on page 10, line 289 and page 11, line 326, respectively.

Response 1: We unified the nomenclature for monoethanolamine.

Point 2:  Page 12, line 366 in equation 2, “theoretical” or “theoretical”?

Response 2: Thank you the remark! The typo was corrected.

Point 3: Page 12, line 373: Did you reduce nitrobenzene or in Conclusion mentioned dinitrotoluene? If 2,4-dinitrobenzene was studied, please, add results for this substrate for comparison of action of newly prepared hydrogenation catalysts.

Response 3: In this study nitrobenzene was hydrogenated. However, we carry out DNT hydrogenation in other experiments. That is why DNT was mentioned accidentally in the conclusion.  

Point 4: Could you mention the work-up and analytical method used for evaluation of hydrogenation results?

Response 4: The following text was inserted in the Experimental section.

˝After hydrogenation studies, quantitative analysis of the samples was performed using an Agilent 7890A gas chromatograph coupled with Agilent 5975C mass selective detector and RTX-624 column (60 m × 0.25 mm × 1.4 μm). The volume of injected sample was 1 μL at a 200:1 split ratio, and the inlet temperature was set to 473 K. Helium was used as a carrier gas (2.28 ml/min) and the oven temperature was set at 323 K for 3 min, then heated to 523 K at a heating rate of 10 K/min and held for a further 3 min. Analytical standards for the main product, by-products and intermediates were obtained from Sigma Aldrich (Burlington, MA, USA) and Dr. Ehrenstorfer Ltd. (Wesel, Nordrhein-Westfalen, Germany).˝

Round 2

Reviewer 2 Report

In present form paper could be published as it is, without additional action from authors